

# Alleviation of dextran sulfate sodium (DSS)-induced colitis in mice through the antioxidative effects of muscone *via* the MyD88/p38 MAPK signalling pathway

Gang Yao[1], Jian Zhang[1], Lingyan Zhang[1], Hai Zhao[1], Shuguang Wu[1], Hongmei Yang[2] and Jiangwei Yu[3]

[1] Institute of Laboratory Animal Science, Guizhou University of Traditional Chinese Medicine, Guiyang, Guizhou, China
[2] The First Affiliated Hospital, Guizhou University of Traditional Chinese Medicine, Guiyang, Guizhou, China
[3] School of Basic Medicine, Guizhou University of Traditional Chinese Medicine, Guiyang, Guizhou, China

Corresponding author
Gang Yao, yaogang901@163.com

## ABSTRACT

**Background**. Inflammatory bowel disease (IBD) is characterized by chronic inflammation. Ulcerative colitis (UC) is a subtype of IBD. The symptoms of UC include inflammation, damage to crypts, and ulceration. UC patients frequently experience comorbid psychological disorders. Muscone has notable anti-inflammatory, antioxidative, and antidementia properties. Our study aimed to investigate the potential of muscone to alleviate colitis, the underlying mechanisms, and the signalling pathways involved.

**Methods**. C57BL/6 mice were administered dextran sulfate sodium (DSS) to induce colitis. The SMART v3.0 (Panlab, Barcelona, Spain) was used to measure parameters in the open field test and the tail suspension test to assess depression and anxiety. Gait changes were analysed using the DigiGait™ imaging system. The severity of colitis was assessed through body weight loss, stool consistency, gross bleeding, and histopathological evaluation. Proteins related to anti-inflammatory and antioxidative effects were analysed in dissociated tissues using mouse-specific commercial enzyme-linked immunosorbent assay (ELISA) kits.

**Results**. Muscone treatment reduced gross bleeding and histopathological damage scores and increased the ratio of colon length to body weight. Gait analysis revealed improvements in swing time, brake time, propulsive time, stance duration, stride duration, stride length, stride frequency, and paw area with muscone treatment. However, muscone treatment did not improve the distance travelled or the time spent in the open field test, nor did it affect the immobility duration in the tail suspension test. We observed that the expression of claudin-1, occludin, and zonula occludens-1 (ZO-1) increased in response to treatment with muscone. Muscone treatment downregulated the expression of interleukin-1β (IL-1β), interleukin-6 (IL-6), interleukin-17 (IL-17), interleukin-33 (IL-33), and tumour necrosis factor-α (TNF-α), while increasing the expression of interleukin-4 (IL-4) and interleukin-10 (IL-10). Muscone treatment increased the concentrations of catalase (CAT), superoxide dismutase (SOD), glutathione peroxidase (GSH-PX), and haem oxygenase (HO-1), and suppressed the expression of myeloperoxidase (MPO), cyclooxygenase-2 (COX-2), and nicotinamide adenine dinucleotide phosphate (NADPH) oxidases (NOX1 and NOX2).

Furthermore, muscone treatment inhibited the expression of myeloid differentiation primary response 88 (MyD88) and p38 mitogen-activated protein kinase (MAPK). **Conclusion**. Muscone effectively alleviated the symptoms of colitis, which may be due to the reduction in reactive oxygen species (ROS). The potential mechanism underlying the mitigation of colitis may involve the inhibition of the MyD88/p38 MAPK signalling pathway. Our studies suggest that muscone could be a promising target for treating IBD in clinical studies.

## INTRODUCTION

Inflammatory bowel disease (IBD) is characterized by chronic inflammation in the lower digestive tract (*Alsararatee, 2024*; *Ismail et al., 2025*). The common symptoms of IBD include abdominal pain, diarrhoea, weight loss, and sometimes colon bleeding (*Alsararatee, 2024*; *Wangchuk, Yeshi & Loukas, 2024*). Patients suffering from IBD experience significant discomfort and a diminished quality of life (*Khan et al., 2024*; *Roderburg et al., 2024*). Ulcerative colitis (UC) is a subtype of IBD that primarily affects the colon (*Wangchuk, Yeshi & Loukas, 2024*). The disease affects males and females in nearly equal proportions (*Blüthner et al., 2024*).

Patients suffering from IBD are at a greater risk of experiencing depression and anxiety than the general population is (*Roderburg et al., 2024*). One previous study revealed that patients with UC exhibit significantly more symptoms of anxiety and depression (*Wang et al., 2024*). Research on these mental disorders has focused predominantly on behavioural indices observed in animal models. Two primary methods utilized for behavioural tests are the open field test and the tail suspension test (*Gencturk & Unal, 2024*). Studies have shown that animals suffering from depression have reduced locomotor activity, number of entries, and time spent in the central zone in the open field test (*Xia et al., 2023*). Compared with control mice, depressed mice exhibit significantly greater immobility in the tail suspension test (*de Lima et al., 2023*). Furthermore, gait analysis constitutes another valuable research method in animal behaviour studies for evaluating the severity of diseases (*Fryer et al., 2024*; *Ren et al., 2024*; *Tueth et al., 2024*). Studies have shown that gait parameters differ between the forelimbs or hind limbs of intact mice (*Leblond et al., 2003*). Therefore, one side of the limbs has been used for gait analysis in some studies. Gait analysis of the left forelimb and hind limb indicated that parameters such as stance, stride duration, forelimb stance width, stride frequency, and gait symmetry are significantly altered in neuroinflammation-mediated brain injury (*Fryer et al., 2024*).

The potential mechanism underlying depressive-like behaviour in individuals has been attributed to oxidative stress, a condition frequently observed in IBD patients (*Ji et al., 2023*; *Pan et al., 2025*). Studies suggest that as patients recover from depression, the levels of oxidative stress-related products are significantly reduced (*Ait Tayeb et al., 2023*). Research has demonstrated that depressive-like behaviour can be mitigated through the

downregulation of the myeloid differentiation primary response 88 (MyD88) and nuclear factor-κB (NF-κB) pathways (*Yang et al., 2019*; *He et al., 2020*).

The common pathological features of UC include tissue destruction, impairment of the gut epithelial barrier, oxidative stress, and inflammation (*Li et al., 2024b*; *Ismail et al., 2025*). Tissue destruction, a hallmark of UC pathology, primarily results from activated apoptosis. Studies have indicated that proapoptotic proteins, such as caspase-3, are significantly upregulated in dextran sulfate sodium (DSS)-induced colitis (*Abd-Ellatieff et al., 2024*). Furthermore, the expression levels of key epithelial barrier proteins, including claudin-1, zonula occludens-1 (ZO-1), and occludin, are significantly decreased in individuals with colitis (*Li et al., 2024d*; *Arumugam, Saha & Nighot, 2025*). Epithelial barrier dysfunction contributes to a cascade of inflammatory responses (*Li et al., 2024d*). Oxidative stress in colitis is characterized by the generation of reactive oxygen species (ROS), which directly promote the production of proinflammatory cytokines such as tumour necrosis factor-α (TNF-α), interleukin-1β (IL-1β), interleukin-6 (IL-6), interleukin-17 (IL-17), and interleukin-33 (IL-33) (*Lin et al., 2023*). Elevated expression levels of these cytokines further exacerbate the inflammatory processes in colitis (*Kałuzna, Olczyk & Komosińska-Vassev, 2022*).

The inflammatory process in colitis involves a cascade of interactions among several crucial proteins, with MyD88 serving as a central player (*Deguine & Barton, 2014*; *Baek et al., 2023*; *Zheng et al., 2024b*). MyD88 receives extracellular signals and subsequently activates downstream signalling pathways (*Deguine & Barton, 2014*). Numerous studies have reported that colitis upregulates MyD88 levels, thereby triggering signalling cascades involved in apoptosis and the mitogen-activated protein kinase (MAPK) and NF-κB pathways (*Abdelzaher et al., 2023*; *Miao et al., 2024*; *Wei et al., 2024*). The activation of these pathways accelerates the inflammatory process, exacerbating the progression of colitis. Conversely, inhibiting the activation of these proteins has been shown to alleviate colitis symptoms. Research has demonstrated that targeting MyD88 inhibition and blocking downstream apoptosis can effectively mitigate the severity of intestinal inflammation (*Abdelzaher et al., 2023*; *Wei et al., 2024*). Additionally, studies indicate that inflammation and oxidative stress in IBD can be ameliorated by modulating the TLR4/MAPK and Keap1/Nrf2 pathways (*Yang et al., 2024*; *Elgohary, Omara & Salama, 2025*). ERK activation has been shown to increase interleukin-10 (IL-10) secretion and alleviate DSS-induced colitis by promoting ERK phosphorylation (*Zheng et al., 2021*; *Qu et al., 2024*). Furthermore, NF-κB represents a classic pathway implicated in inflammation progression, and inhibiting its activity effectively prevents inflammatory development, significantly reducing the severity of IBD (*Baek et al., 2023*; *Ismail et al., 2025*).

Musk has been incorporated into more than 400 traditional Chinese medicine formulations, primarily because of its pharmacological properties that affect inflammation, immune system disorders, and neurological conditions (*Liu et al., 2023*; *Yang et al., 2023*; *Li, Zhuang & Jiang, 2024c*). Muscone is a primary active component of musk, and its chemical structure is 3-methylcyclopentadecanone (*Zhu et al., 2022*). Muscone has notable anti-inflammatory, antioxidative, antidementia, and anticerebral ischaemic properties, as evidenced by various studies (*Wang et al., 2020*; *Liu et al., 2022*; *Liu et al., 2023*; *Zhang et al.,*

*2023*). Muscone ameliorates oxidative stress by reducing ROS generation and enhancing the activity of endogenous antioxidative enzymes, including haem oxygenase (HO-1), superoxide dismutase (SOD), and glutathione peroxidase (GSH-PX) (*Yu et al., 2014*; *Phung et al., 2020*; *Huang et al., 2025*). The administration of muscone has been shown to decrease the expression levels of TNF-α, IL-1β, IL-6, interleukin-8 (IL-8), IL-17, and interleukin-18 (IL-18) (*Wang et al., 2020*; *Liu et al., 2023*; *Li, Zhuang & Jiang, 2024c*). This regulatory effect on cytokine expression highlights the therapeutic potential of muscone in the treatment of inflammatory diseases (*Zhou et al., 2020*). Furthermore, studies indicate that muscone mitigates inflammation through modulation of the NF-κB and NLRP3/IL-1β /p38 MAPK pathways (*Zhai et al., 2020*; *Yang et al., 2023*; *Li, Zhuang & Jiang, 2024c*).

IBD patients with depressive and anxiety behaviours often fail to receive effective treatment, and research addressing these conditions remains limited. Our work examined the alleviating effects of muscone on IBD from both behavioural and conventional pathological perspectives. The objectives of this study were to investigate: (1) the potential of muscone to alleviate inflammation in DSS-induced colitis; (2) the mechanisms underlying its anti-inflammatory effects; and (3) the signalling pathways involved in the mitigation of colitis symptoms.

## MATERIALS & METHODS

### Animals

C57BL/6 mice were bred at the Institute of Laboratory Animal Science, Guizhou University of Traditional Chinese Medicine. Both male and female mice are susceptible to developing severe UC when induced with DSS (*Chassaing et al., 2014*). An equal number of male and female mice, each weighing $20 \pm 2$ g, were used in the study. The mice were housed in standard cages, with five mice per cage, and maintained in a controlled environment at a temperature of $22 \pm 2$ °C and humidity of $55 \pm 5$%. A 12-hour light/dark cycle was maintained. The mice were provided with a complete nutritional compound diet and had ad libitum access to foraging food. A one-week acclimatization period was allowed before the experiment.

The animal experiments were performed for a total of 16 days, with the first seven days allocated to the induction of colitis and the subsequent nine days allocated to drug treatment. After nine days of treatment, the mice were euthanized by cervical dislocation under isoflurane anaesthesia. The euthanasia procedures were consistent with those in other studies (*Li, Lin & Su, 2024a*). The colon length of the mice was measured, and the ratio of colon length to body weight was calculated. A segment of fresh colonic tissue was excised for histopathological analysis and fixed in 4% paraformaldehyde for more than 24 h. The remaining colonic tissue was harvested and stored at $-80$ °C for subsequent enzyme-linked immunosorbent assay (ELISA) analysis. Data from animals that died during the experimental period were excluded.

The experimental protocols were approved by the Animal Experimental Ethical Committee of Guizhou University of Traditional Chinese Medicine (No. 20230062).

## DSS-induced colitis

We determined the sample sizes on the basis of previous studies (*Du et al., 2018*; *Festing, 2018*; *He et al., 2020*; *Liu et al., 2022*). Thirty-eight mice were randomly allocated into three groups: a control group ($n = 12$), a model group ($n = 13$), and a muscone group ($n = 13$). The UC mouse model was induced using 2.5% (w/v) DSS (MW: 36,000–50,000) (*Lai et al., 2024*). The muscone group received DSS drinking water for the first seven days, followed by normal drinking water thereafter. In addition, the muscone group was administered muscone (10 mg/kg; CAS: 541-91-3, a synthetic compound purchased from Macklin Inc., Shanghai, China) daily by gavage after the withdrawal of DSS drinking water (*Liang et al., 2010*). Muscone was prepared as a 1 g/L emulsion using 1% Tween 80 in normal saline (*Hui et al., 2003*). The emulsion volume was adjusted to ensure a gavage dose of 10 mg/kg per mouse. The control group was provided with DSS-free drinking water throughout the study and received normal saline gavage daily after the first seven days. The model group was initially given DSS drinking water for seven days, after which they were switched to normal drinking water with no additional treatment except for normal saline gavage every day.

## Behavior evaluation

To assess symptoms of comorbid depression and anxiety, SMART v3.0 (Panlab, Barcelona, Spain) was used to measure various parameters during the open field test, including total distance travelled; distance in the center, border, and periphery of the arena; and duration. The open field arena was cleaned with 75% ethanol before each mouse was tested. After ethanol volatilization, the mouse was placed in the centre of the arena for one minute of acclimation, followed by a five-minute video recording.

SMART v3.0 (Panlab) was also used to measure the total immobility time and the mean duration of immobility during the tail suspension test. The mice were suspended by their tails using adhesive tape in a rectangular box equipped with a hook, while a side-mounted camera recorded a five-minute video. The rectangular box was cleaned with 75% ethanol, and the next mouse was tested after ethanol volatilization.

A DigiGait™ imaging system (Mouse Specifics, Inc, Framingham, MA, USA) was used to analyse the gait changes in the mice. The gait parameters mainly included swing duration, brake time, propulsive time, stance duration, stride duration, stride length, stride frequency, and paw area. Each mouse was placed on the treadmill belt for acclimation before the experiment, and mice that failed to run on the belt were excluded. A two-minute video was recorded once the velocity of the mouse was consistent with that of the treadmill belt. The treadmill belt was cleaned with 75% ethanol before each mouse experiment.

## Assessment of body weight loss, stool consistency, and colonic bleeding

The severity of clinical colitis was assessed on the basis of body weight loss, stool consistency, and gross bleeding (*Pervin et al., 2016*). Body weight loss was calculated as the percentage difference between the initial body weight after acclimation and the body weight on a specific day. Body weight loss was categorized into four scores: 0 (none), 1 (1–4.99%), 2

(5–10%), and 3 (>10%). Stool consistency was graded on a scale of four scores: 0 (normal), 1 (slightly loose faeces), 2 (loose faeces), and 3 (watery diarrhoea). Gross bleeding was assessed by evaluating faecal blood content with four rating scores: 0 (normal), 1 (slightly bloody), 2 (bloody), and 3 (blood present throughout the stool).

## Histopathological evaluation

Paraffin-embedded sections were prepared to assess damage to the colonic epithelial tissue. The tissue samples fixed in 4% paraformaldehyde underwent a systematic procedure, which included dehydration, clearing, paraffin embedding, sectioning, and staining with haematoxylin and eosin. The integrity of the epithelial tissue was evaluated on the basis of damage to goblet cells and crypts, using a four-point scoring system: 0 (no damage), 1 (mild damage to goblet cells), 2 (extensive damage to goblet cells or mild damage to crypts), and 3 (extensive damage to crypts or infiltration of inflammatory cells in the basal mucus region).

## Expression levels of proteins related to anti-inflammation and antioxidation

ELISA was used to analyse protein expression levels in our study. The analysis was conducted using a mouse-specific commercial ELISA kit (Shanghai Enzyme Link Biotechnology Co., Ltd., Shanghai, China). The experimental procedures were carried out according to the manufacturer's instructions. Briefly, a 0.1 g sample was homogenized with 0.9 mL of PBS (pH 7.4) using a homogenizer. The homogenate was centrifuged at 2,500 revolutions per minute (rpm) for 20 min, after which the supernatant was collected. A microplate reader (RT-6100, Rayto Life and Analytical Sciences Co. Ltd., Shenzhen, China) was used to quantify the target protein concentration in the supernatant, which was then converted to reflect its tissue content on the basis of sample weight.

The expression levels of claudin-1, occludin, and ZO-1 were measured using an ELISA kit to assess whether muscone improved intestinal epithelial barrier function. Additionally, the level of mucin 2 (MUC-2) in the lumen was measured to assess intestinal barrier integrity.

The expression levels of proteins associated with anti-inflammatory and antioxidative effects were measured using ELISA kits to assess the anti-inflammatory and antioxidative activities of muscone. The assessed inflammatory cytokines included IL-1β, IL-6, TNF-α, IL-17, IL-4, IL-10, and IL-33. The proteins related to oxidative damage measured in our study included inducible nitric oxide synthase (iNOS), NAD (P) H: quinone oxidoreductase 1 (NQO1), catalase (CAT), SOD, GSH-PX, HO-1, cyclooxygenase-2 (COX-2), myeloperoxidase (MPO), NOX1, and NOX2.

To investigate the anti-inflammatory and antioxidative mechanisms of muscone, the expression levels of key proteins in signalling pathways, including toll-like receptor 4 (TLR4), p38 MAPK, phospho-p38 MAPK (p-p38 MAPK), Keap1, Nrf2, Caspase-3, MyD88, ERK, phospho-ERK (p-ERK), and NF-κB, were measured.

## Statistical analysis

The outliers in the dataset were identified utilizing the isolation forest algorithm and subsequently replaced with the average value of the group. The data are reported as the means ± standard deviations (means ± SDs). Independent samples t-tests were used to compare the means between two groups, whereas one-way ANOVA was used to analyse multiple comparisons among groups of three or more. If the variances of the groups were equal, we compared multiple groups using the least significant difference (LSD) method; if not, we utilized Tamhane's T2 method for comparison. Kaplan–Meier survival curves were generated to analyse variations in animal survival, and tests for trends were designed to detect ordered differences using a log-rank test. The statistical significance level was set at $p < 0.05$. All the statistical analyses were performed using R software (*R Core Team, 2023*).

# RESULTS

## Analysis of DSS-induced colitis symptoms

The effects of muscone on DSS-induced colitis are displayed in Fig. 1. Our results indicated that the mice experienced body weight loss after receiving DSS in their drinking water, with a slight recovery upon DSS removal (Fig. 1B). The bloody stools were observed three days after the administration of DSS (Fig. 1C). The scores for bloody stools were slightly lower in muscone-treated mice than in the model group. The survival probability of the model group was significantly lower than that of the muscone group (Fig. 1D). The ratio of colon length to body weight was significantly greater in the muscone group than in the model group at the end of the study (Fig. 1E). The histological scores are presented in Fig. 1F. Histological analysis revealed significantly more severe damage in the model group (Figs. 1G–1I). Many goblet cells and crypts were missing in the model group, with inflammatory cell infiltration observed at the base of the mucosa. The damage to epithelial cells was alleviated in the muscone-treated group.

## Analysis of behaviour
### Open field test
The open field test indicated no significant difference in the total distance travelled between the model and muscone groups. No significant differences were observed in the distances travelled in the centre, border, or peripheral zones between the model and muscone groups (Fig. 2). Similarly, no significant differences were found in the duration spent in the centre, border, or peripheral zones (Fig. 3).

### Tail suspension test
The tail suspension test revealed no difference in the total immobility duration between the model and muscone groups (Fig. 4). The mean immobility duration in the muscone group was not significantly different from that in the model group.

### Gait analysis
Gait analysis revealed significant differences between the model and muscone groups (Fig. 5, Table 1). The swing time of the left hind limb was significantly longer in the model

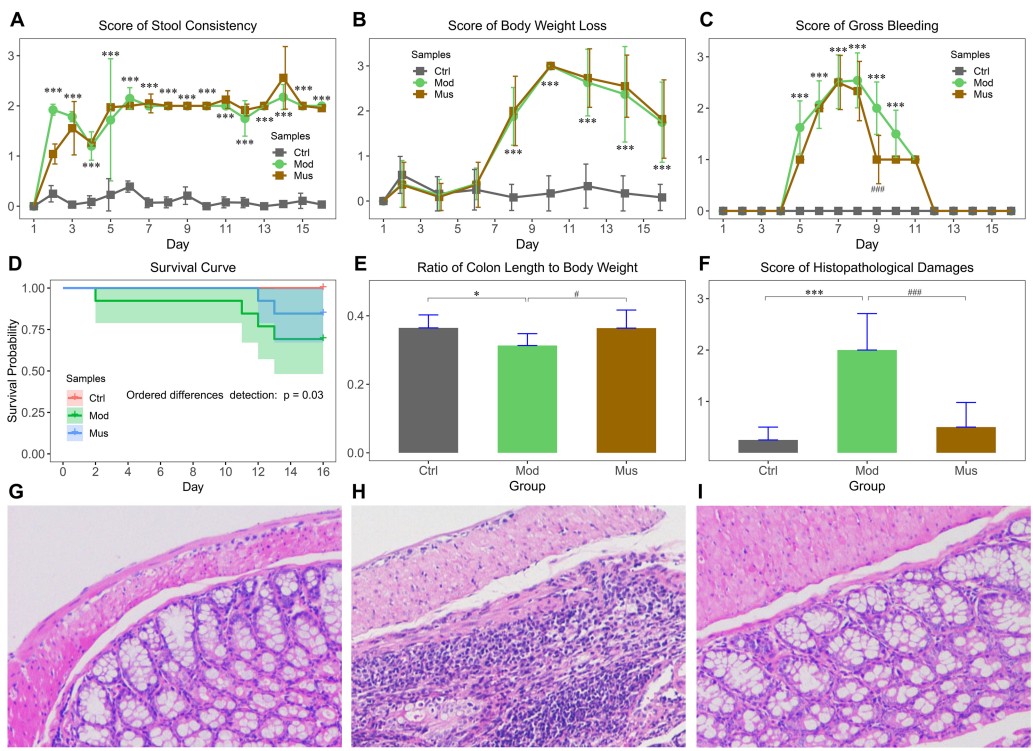

**Figure 1 Morphological characteristics of DSS-induced colitis.** (A) Average stool consistency scores during the study. (B) Average body weight loss scores during the study. (C) Average gross bleeding scores decreased after muscone treatment. (D) Ordered differences detected using a log-rank test revealed significant differences among groups in the survival curve. (E) The ratio of colon length to body weight improved with muscone treatment. (F) Average histopathological damage scores decreased after muscone treatment. (G, H, and I) show the histological features of the mice in the control, model, and muscone groups. Ctrl, control group; Mod, model group; Mus, muscone group. The data in (A, B, C, E, and F) are reported as the means ± SDs. Independent samples $t$-test between the control group ($n = 12$) and the model group ($n = 9$): * $p < 0.05$; ** $p < 0.01$, *** $p < 0.001$. Independent samples $t$-test between the model group ($n = 9$) and the muscone group ($n = 11$): # $p < 0.05$; ## $p < 0.01$, ### $p < 0.001$.

group than in the muscone group. The brake time of the right hind limb was longer in the model group than in the muscone group. The propulsive time of the hind limbs and the right forelimb was longer in the model group than in the muscone group. The stance duration of all limbs was longer in the model group than in the muscone group. The stride duration of the left forelimb and hind limbs was longer in the model group than in the muscone group. The stride lengths of all four limbs in the model group were significantly shorter than those in the muscone group. The stride frequency of the left forelimb and hind limbs in the model group was lower than that in the muscone group. The paw area of the hind paws in the model group was larger than that in the muscone group.

## Measurement of intestinal epithelial barrier proteins

The epithelial barrier proteins claudin-1, occludin, and ZO-1 were significantly reduced in the mice treated with DSS (Fig. 6, Table 2). After treatment with muscone, the expression levels of claudin-1, occludin, and ZO-1 were significantly increased. However, the

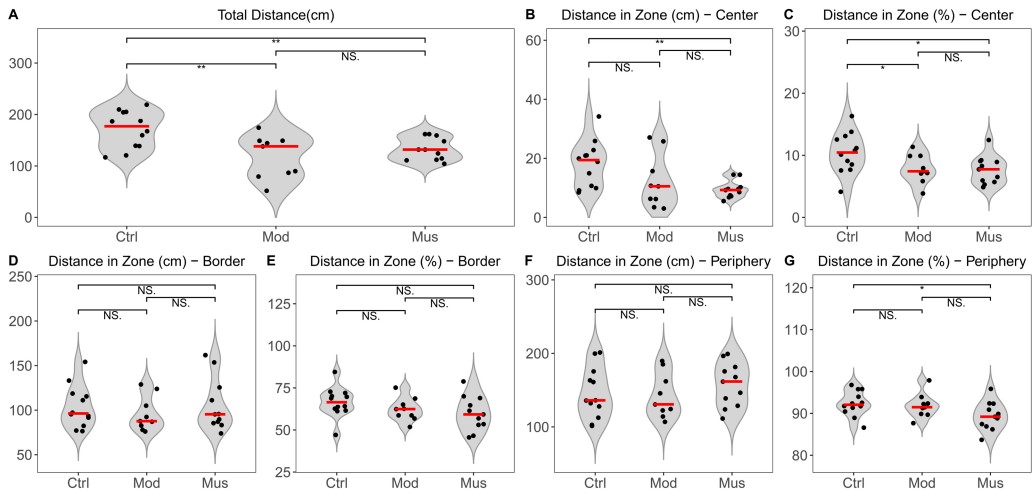

**Figure 2 Distance travelled in the open field test arena.** (A) Total distance travelled in the open field test arena. (B) Distance travelled in the central zone. (C) Percentage of distance travelled in the central zone relative to the total distance. (D) Distance travelled in the border zone. (E) Percentage of distance travelled in the border zone relative to the total distance. (F) Distance travelled in the peripheral zone. (G) Percentage of distance travelled in the peripheral zone relative to the total distance. Ctrl, control group; Mod, model group; Mus, muscone group; NS, non-significant. The open field test was performed after the completion of the muscone treatment, and the results were analysed using SMART v3.0 (Panlab). One-way analysis of variance (ANOVA) was used for comparisons among the control group ($n = 12$), model group ($n = 9$), and muscone group ($n = 11$): * $p < 0.05$; ** $p < 0.01$; *** $p < 0.001$.

expression level of MUC-2 in the muscone group was not significantly different from that in the model group.

## Measurement of proinflammatory cytokines

A decrease in the expression levels of several proinflammatory cytokines was observed in mice treated with muscone compared with those in the model group (Fig. 7, Table 2). The concentrations of IL-1β, IL-6, TNF-α, IL-17, and IL-33 were significantly lower in the mice treated with muscone than in those in the model group. Our results revealed significantly higher expression levels of IL-4 and IL-10 in the muscone group than in the model group.

## Evaluation of antioxidant activity

Compared with those in the model group, the antioxidative activity of the mice in the muscone group was significantly greater (Fig. 8, Table 2). The concentrations of CAT, SOD, GSH-PX, and HO-1 were significantly greater in the muscone group than in the model group. The concentrations of MPO, COX-2, NOX1, and NOX2 were significantly lower in the muscone group than in the model group.

## Potential signalling pathways involved in colitis resistance

No significant differences were observed in the concentrations of Keap-1 and Nrf2 between the mice in the model and muscone groups (Fig. 9, Table 2). No differences in the expression levels of TLR4, p-ERK, or NF-κB were detected between the mice in the model and muscone groups. The expression levels of caspase-3, MyD88, p38 MAPK, and

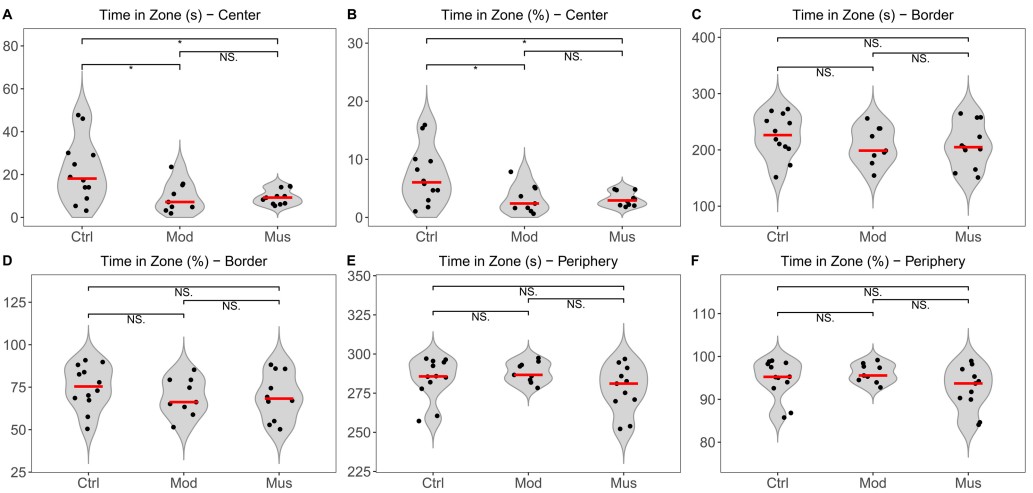

**Figure 3  Time spent in the open field test arena.** (A) Time spent in the central zone. (B) Percentage of time spent in the central zone relative to total time. (C) Time spent in the border zone. (D) Percentage of time spent in the border zone relative to the total time. (E) Time spent in the peripheral zone. (F) Percentage of time spent in the peripheral zone relative to the total time. Ctrl, control group; Mod, model group; Mus, muscone group; NS, non-significant. Videos were recorded and analysed using SMART v3.0 (Panlab) after the completion of the muscone treatment. One-way ANOVA was used for comparisons among the control group ($n = 12$), model group ($n = 9$), and muscone group ($n = 11$): * $p < 0.05$; ** $p < 0.01$, *** $p < 0.001$.

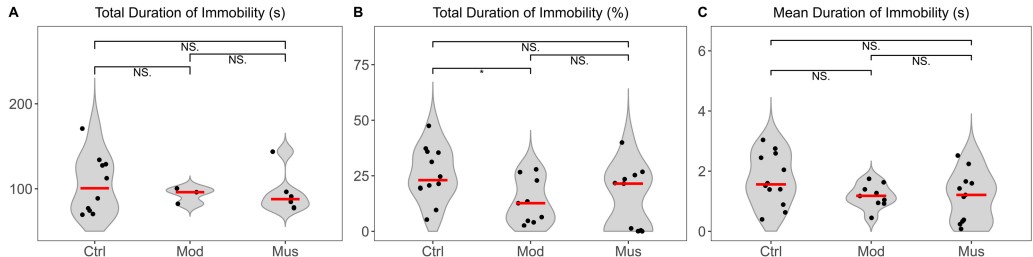

**Figure 4  Immobility duration in the tail suspension test.** (A) Total duration of immobility in the tail suspension test. (B) Percentage of the total duration of immobility relative to the total recorded time. (C) The mean duration of immobility in the tail suspension test. Ctrl, control group; Mod, model group; Mus, muscone group; NS, non-significant. Videos were recorded and analysed using SMART v3.0 (Panlab) after the completion of the muscone treatment. One-way ANOVA was used for comparisons among the control group ($n = 12$), model group ($n = 9$), and muscone group ($n = 11$): * $p < 0.05$; ** $p < 0.01$, *** $p < 0.001$.

p-p38 MAPK were significantly lower in the muscone group than in the model group, whereas the expression of ERK was significantly higher in the muscone group than in the model group (Fig. 10).

## DISCUSSION

DSS-induced colitis is a commonly used experimental model for studying IBD (*Li et al., 2024d*; *Qu et al., 2024*; *Su et al., 2024*). We observed that the mice induced with DSS

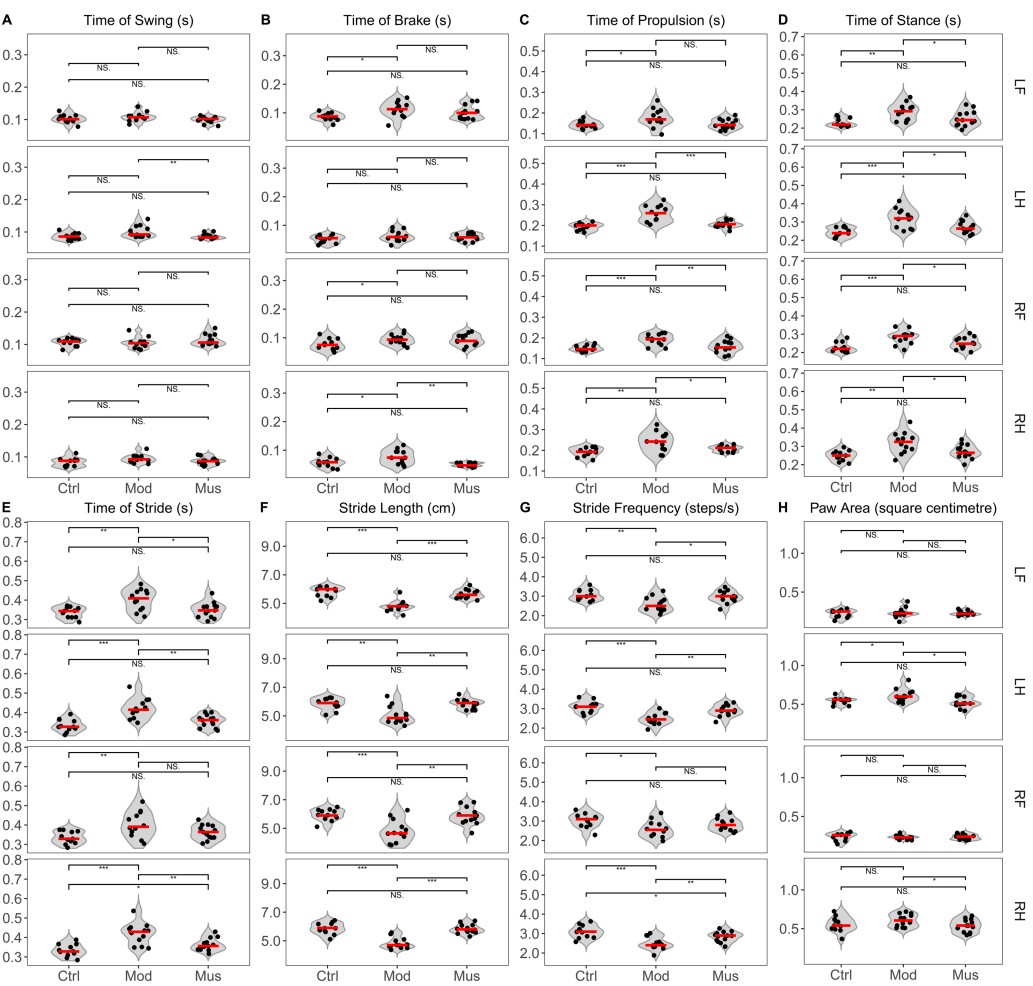

**Figure 5** **Gait parameters of the mice in the study.** (A) The average swing time of the left hind limb was shorter in the muscone group than in the model group. (B) The average brake time of the right hind limb was shorter in the muscone group than in the model group. (C) The average propulsive time of the hind limbs and right forelimb was shorter in the muscone group than in the model group. (D) The average stance time of all limbs was shorter in the muscone group than in the model group. (E) The average stride time of the hind limbs and left forelimb was shorter in the muscone group than in the model group. (F) The average stride length of all limbs was longer in the muscone group than in the model group. (G) The average stride frequency of the left forelimb and hind limbs was greater in the muscone group than in the model group. (H) The average paw area of the hind limbs was smaller in the muscone group than in the model group. LF, left forelimb; LH, left hind limb; RF, right forelimb; RH, right hind limb. Ctrl, control group; Mod, model group; Mus, muscone group; NS, non-significant. Gait analysis was performed after the completion of muscone treatment using the DigiGait™ imaging system. One-way ANOVA was used for comparisons among the control group ($n = 11$), model group ($n = 11$), and muscone group ($n = 13$): * $p < 0.05$; ** $p < 0.01$, *** $p < 0.001$.

experienced increased body weight loss, decreased stool consistency, and increased colon bleeding (Fig. 1). These pathological symptoms indicated the successful establishment of DSS-induced experimental colitis.

Muscone is the primary ingredient of musk that has been extensively used in traditional Chinese medicine for thousands of years (*Zhu et al., 2022*; *Yang et al., 2023*). Our

**Table 1 Results of gait analysis in the control, model, and muscone groups.** The data are reported as the means ± SDs.

| | Limbs | Groups | | |
| --- | --- | --- | --- | --- |
| | | Control | Model | Muscone |
| Time of swing (s) | LF | $0.103 \pm 0.013$ | $0.108 \pm 0.014$ | $0.099 \pm 0.010$ |
| | RF | $0.106 \pm 0.012$ | $0.105 \pm 0.017$ | $0.113 \pm 0.017$ |
| | LH | $0.087 \pm 0.010$ | $0.101 \pm 0.018$ | $0.086 \pm 0.008^{\#}$ |
| | RH | $0.084 \pm 0.014$ | $0.094 \pm 0.013$ | $0.087 \pm 0.011$ |
| Time of brakes (s) | LF | $0.087 \pm 0.014^{*}$ | $0.113 \pm 0.027$ | $0.101 \pm 0.024$ |
| | RF | $0.077 \pm 0.020^{*}$ | $0.094 \pm 0.018$ | $0.092 \pm 0.020$ |
| | LH | $0.052 \pm 0.013$ | $0.062 \pm 0.019$ | $0.059 \pm 0.012$ |
| | RH | $0.057 \pm 0.017^{*}$ | $0.077 \pm 0.025$ | $0.050 \pm 0.007^{\#\#}$ |
| Time of propulsion (s) | LF | $0.144 \pm 0.018^{*}$ | $0.177 \pm 0.045$ | $0.147 \pm 0.024$ |
| | RF | $0.150 \pm 0.015^{***}$ | $0.191 \pm 0.027$ | $0.158 \pm 0.030^{\#\#}$ |
| | LH | $0.194 \pm 0.017^{***}$ | $0.258 \pm 0.039$ | $0.207 \pm 0.017^{\#\#\#}$ |
| | RH | $0.192 \pm 0.022^{**}$ | $0.243 \pm 0.047$ | $0.208 \pm 0.014^{\#}$ |
| Time of stance (s) | LF | $0.231 \pm 0.023^{**}$ | $0.290 \pm 0.045$ | $0.253 \pm 0.042^{\#}$ |
| | RF | $0.227 \pm 0.027^{***}$ | $0.283 \pm 0.038$ | $0.250 \pm 0.031^{\#}$ |
| | LH | $0.246 \pm 0.025^{***}$ | $0.320 \pm 0.052$ | $0.271 \pm 0.033^{\#}$ |
| | RH | $0.249 \pm 0.028^{**}$ | $0.320 \pm 0.058$ | $0.271 \pm 0.037^{\#}$ |
| Time of stride (s) | LF | $0.337 \pm 0.028^{**}$ | $0.402 \pm 0.054$ | $0.349 \pm 0.040^{\#}$ |
| | RF | $0.335 \pm 0.032^{**}$ | $0.403 \pm 0.066$ | $0.364 \pm 0.037$ |
| | LH | $0.334 \pm 0.032^{***}$ | $0.421 \pm 0.052$ | $0.357 \pm 0.032^{\#\#}$ |
| | RH | $0.333 \pm 0.032^{***}$ | $0.419 \pm 0.056$ | $0.363 \pm 0.032^{\#\#}$ |
| Stride length (cm) | LF | $5.836 \pm 0.353^{***}$ | $4.808 \pm 0.390$ | $5.66\,2 \pm 0.307^{\#\#\#}$ |
| | RF | $5.945 \pm 0.411^{***}$ | $4.825 \pm 0.801$ | $5.869 \pm 0.595^{\#\#}$ |
| | LH | $5.855 \pm 0.413^{**}$ | $5.042 \pm 0.630$ | $5.838 \pm 0.328^{\#\#}$ |
| | RH | $5.855 \pm 0.403^{***}$ | $4.825 \pm 0.393$ | $5.831 \pm 0.320^{\#\#\#}$ |
| Stride frequency(steps/s) | LF | $3.064 \pm 0.280^{**}$ | $2.583 \pm 0.381$ | $2.962 \pm 0.320^{\#}$ |
| | RF | $3.036 \pm 0.367^{*}$ | $2.583 \pm 0.426$ | $2.838 \pm 0.312$ |
| | LH | $3.091 \pm 0.302^{***}$ | $2.467 \pm 0.306$ | $2.892 \pm 0.290^{\#\#}$ |
| | RH | $3.100 \pm 0.313^{***}$ | $2.483 \pm 0.321$ | $2.838 \pm 0.272^{\#\#}$ |
| Paw area(square centimetre) | LF | $0.222 \pm 0.050$ | $0.233 \pm 0.065$ | $0.228 \pm 0.028$ |
| | RF | $0.236 \pm 0.051$ | $0.232 \pm 0.027$ | $0.235 \pm 0.033$ |
| | LH | $0.545 \pm 0.045^{*}$ | $0.612 \pm 0.084$ | $0.526 \pm 0.066^{\#}$ |
| | RH | $0.555 \pm 0.098$ | $0.607 \pm 0.080$ | $0.536 \pm 0.083^{\#}$ |

**Notes.**

LF, left forelimb; RF, right forelimb; LH, left hind limb; RH, right hind limb.

Statistical significance is indicated as follows:

$^{*}p < 0.05$.

$^{**}p < 0.01$.

$^{***}p < 0.001$ (comparison between the control and model groups).

$^{\#}p < 0.05$.

$^{\#\#}p < 0.01$.

$^{\#\#\#}p < 0.001$ (comparison between the muscone and model groups).

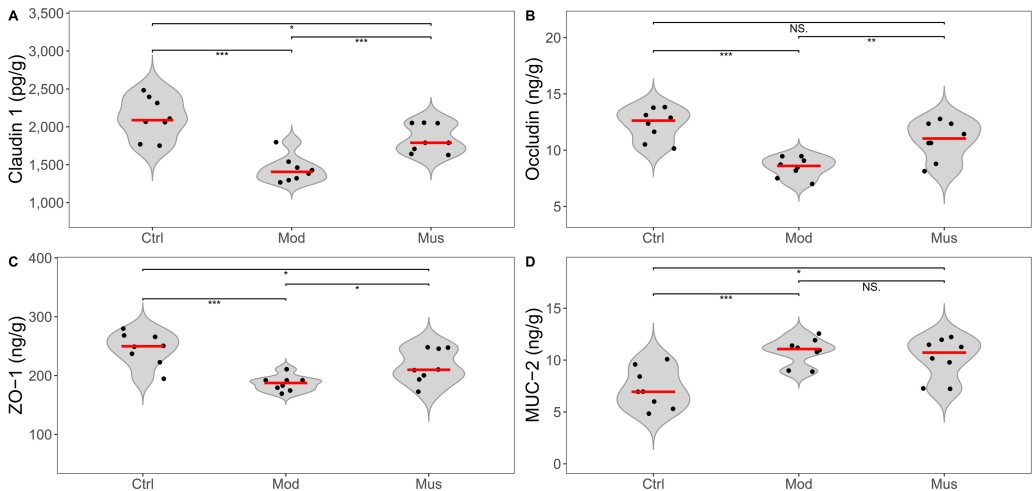

**Figure 6 Expression levels of intestinal epithelial barrier proteins.** (A) Claudin 1 expression was greater in the muscone group than in the model group after muscone treatment. (B) Occludin expression was greater in the muscone group than in the model group after muscone treatment. (C) ZO-1 expression was greater in the muscone group than in the model group after muscone treatment. (D) No difference in MUC-2 expression was detected between the model group and the muscone group. Ctrl, control group; Mod, model group; Mus, muscone group; NS, non-significant. Fresh colon tissue was harvested after the completion of the muscone treatment. Epithelial barrier proteins were quantified using ELISA after the dissociation of colon tissue. One-way ANOVA was used to compare the epithelial barrier proteins among the control group ($n = 8$), model group ($n = 8$), and muscone group ($n = 8$): * $p < 0.05$; ** $p < 0.01$, *** $p < 0.001$.

observations indicated that the mice treated with muscone presented less colon bleeding than did the untreated mice (Fig. 1C). Histological analysis revealed that the colon tissue of muscone-treated mice recovered significantly better than did that of untreated mice (Figs. 1F–1I). The ratio of colon length to body weight was significantly greater in treated mice than in untreated mice (Fig. 1E). These findings demonstrate the effective alleviation of experimental colitis by muscone. In traditional Chinese medicine, muscone is used for treating digestive system disorders (*Zhou et al., 2022*). Studies have demonstrated that patients with anal fissures treated with medications primarily containing musk experience significant alleviation of pain and bleeding symptoms (*Liu, Liu & Jiang, 2017*). These results indicate that muscone has therapeutic potential in the treatment of disorders affecting the posterior or terminal regions of the gastrointestinal tract.

Depression and anxiety are frequently comorbid mental disorders in IBD patients, as reported in previous studies (*Roderburg et al., 2024*; *Wang et al., 2024*). In animal studies, depression and anxiety are typically assessed through behavioural measurements (*Gencturk & Unal, 2024*). Depression is frequently assessed using measures such as distance travelled and time spent in the open field test, whereas anxiety is assessed by immobility time in the tail suspension test (*Kremer et al., 2021*). Studies have shown that muscone can ameliorate lipopolysaccharides (LPS)-induced depression-like behaviour in animal models (*He et al., 2020*). However, we did not observe any differences in distance travelled or time spent in the open field arena (Figs. 2–3) or immobility duration in the tail suspension

**Table 2** **Expression levels of proteins related to anti-inflammation and antioxidation.** The data are reported as the means ± SDs.

| | Groups | | |
|---|---|---|---|
| | **Control** | **Model** | **Muscone** |
| Claudin 1 (pg/g) | 2119.30 ± 269.75[***] | 1437.23 ± 171.71 | 1749.75 ± 213.12[###] |
| Occludin (ng/g) | 12.29 ± 1.40[***] | 8.49 ± 0.89 | 10.89 ± 1.70[##] |
| ZO-1 (ng/g) | 246.06 ± 27.59[***] | 186.68 ± 13.01 | 205.80 ± 42.22[#] |
| MUC-2 (ng/g) | 7.27 ± 1.93[***] | 10.18 ± 1.98 | 10.84 ± 1.30 |
| IL-1β (pg/g) | 145.52 ± 33.47[*] | 188.19 ± 29.19 | 159.50 ± 19.67[#] |
| IL-6 (pg/g) | 126.65 ± 29.13[***] | 195.35 ± 23.84 | 167.39 ± 24.23[#] |
| TNF-α (pg/g) | 425.19 ± 56.73[*] | 493.40 ± 26.22 | 337.79 ± 82.61[###] |
| IL-17 (pg/g) | 50.25 ± 8.99[***] | 72.02 ± 9.31 | 62.38 ± 8.61[#] |
| IL-4 (pg/g) | 235.03 ± 26.38[***] | 175.60 ± 27.85 | 215.54 ± 23.07[##] |
| IL-33 (pg/g) | 142.93 ± 24.88[***] | 249.93 ± 32.32 | 182.61 ± 31.41[###] |
| IL-10 (pg/g) | 711.32 ± 88.71[***] | 527.87 ± 78.71 | 610.49 ± 94.85[#] |
| CAT (pg/g) | 1411.54 ± 188.73[**] | 1021.17 ± 200.66 | 1230.90 ± 176.64[#] |
| SOD (ng/g) | 149.93 ± 18.10[***] | 86.26 ± 21.19 | 134.19 ± 13.32[###] |
| GSH-PX (ng/g) | 271.06 ± 39.90[***] | 193.79 ± 35.15 | 235.31 ± 29.78[#] |
| HO-1 (U/g) | 0.098 ± 0.011[***] | 0.068 ± 0.082 | 0.081 ± 0.013[#] |
| iNOS (ng/g) | 24.11 ± 5.05[**] | 31.41 ± 2.98 | 27.58 ± 6.10 |
| MPO (pg/g) | 2699.22 ± 183.24[***] | 3660.67 ± 414.46 | 2788.63 ± 339.34[###] |
| COX-2 (ng/g) | 60.37 ± 16.31[**] | 90.55 ± 12.19 | 70.42 ± 14.16[##] |
| NOX1 (µg/g) | 58.72 ± 8.50[***] | 111.18 ± 15.26 | 78.57 ± 14.21[###] |
| NOX2 (ng/g) | 76.41 ± 14.52[***] | 147.14 ± 15.38 | 95.30 ± 23.13[###] |
| NQO1 (ng/g) | 30.58 ± 3.38[***] | 18.82 ± 3.41 | 20.60 ± 3.77 |
| Caspase-3 (pg/g) | 0.083 ± 0.012[**] | 0.102 ± 0.006 | 0.092 ± 0.008[#] |
| MyD88 (pg/g) | 2753.16 ± 470.66[**] | 3664.39 ± 515.23 | 2860.91 ± 309.71[##] |
| ERK (pg/g) | 8312.59 ± 762.62[***] | 5887.57 ± 1193.26 | 7172.22 ± 989.11[#] |
| pERK (pg/g) | 3734.95 ± 384.32[***] | 2702.57 ± 380.30 | 2900.20 ± 428.83 |
| TLR4 (ng/g) | 19.50 ± 5.64[**] | 30.10 ± 4.63 | 29.09 ± 5.33 |
| p38 MAPK (ng/g) | 1167.03 ± 175.77[***] | 1783.46 ± 154.72 | 1545.71 ± 183.46[#] |
| p-p38 MAPK (ng/g) | 849.23 ± 167.40[**] | 1104.76 ± 134.00 | 963.32 ± 49.93[#] |
| NF-κB (ng/g) | 4721.92 ± 837.69[**] | 6204.52 ± 500.72 | 6101.87 ± 933.44 |
| Keap1 (pg/g) | 381.91 ± 45.51[***] | 502.76 ± 63.99 | 445.73 ± 68.36 |
| Nrf2 (pg/g) | 1562.62 ± 222.88[**] | 1149.84 ± 249.38 | 1168.37 ± 271.91 |

**Notes.**
Statistical significance is indicated as follows:
[*]$p < 0.05$.
[**]$p < 0.01$.
[***]$p < 0.001$ (comparison between the control and model groups).
[#]$p < 0.05$.
[##]$p < 0.01$.
[###]$p < 0.001$ (comparison between the muscone and model groups).

test (Fig. 4) between the mice that were administered muscone and those that were not. Previous research has demonstrated that drug administration routes can significantly affect therapeutic efficacy (*Magill et al., 2023*). Therefore, the observed discrepancies between our results and previous findings may stem from variations in muscone administration routes.

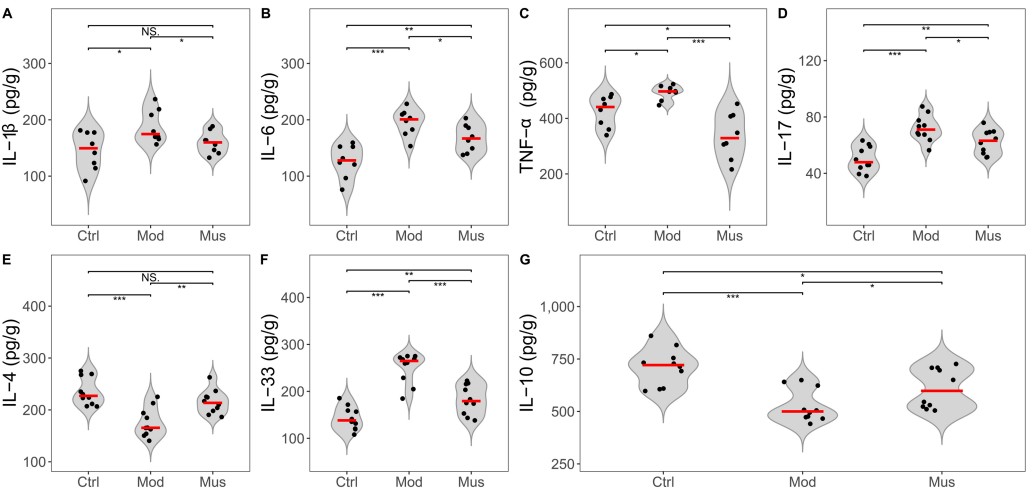

**Figure 7 Expression levels of inflammatory cytokines.** (A) IL-1β expression was lower in the muscone group than in the model group. (B) IL-6 expression was lower in the muscone group than in the model group. (C) TNF-α was lower in the muscone group compared with the model group. (D) IL-17 expression was lower in the muscone group than in the model group. (E) IL-4 expression was greater in the muscone group compared with the model group. (F) IL-33 expression was lower in the muscone group than in the model group. (G) IL-10 expression was greater in the muscone group compared with the model group. Ctrl, control group; Mod, model group; Mus, muscone group; NS, non-significant. Fresh colon tissue was harvested after the completion of the muscone treatment. Inflammatory cytokines were quantified using ELISA after the dissociation of colon tissue. One-way ANOVA was used to compare the inflammatory cytokines among the control group ($n = 8$ or 10), model group ($n = 8$ or 10), and muscone group ($n = 8$ or 10): * $p < 0.05$; ** $p < 0.01$, *** $p < 0.001$.

Rigorous preclinical investigations are needed to systematically evaluate the therapeutic effects of different administration modalities on ulcerative colitis patients with comorbid depressive and anxiety symptoms.

Diseases, chronic pain, and age can all affect gait (*Ben Chaabane et al., 2023*). Gait analysis is a superior method for evaluating pain, inflammatory injury, and locomotor function in patients (*Parent & Moffet, 2003*; *Xu et al., 2019*; *Bonanno et al., 2023*). Individuals with diseases, such as dementia and Parkinson's disease, exhibit altered gait compared with healthy individuals (*Hodgson et al., 2024*). A recent study demonstrated that rats with intraplantar complete Freund's adjuvant-induced inflammatory pain exhibited significant changes in the gait parameters of swing (*Xu et al., 2019*). Our results indicated that the mice treated with muscone presented a significantly shorter stance, shorter stride duration, longer stride length, and greater stride frequency (steps per second) compared to untreated mice (Fig. 5), suggesting that muscone enhances the locomotor ability of mice with DSS-induced colitis. Studies have demonstrated that muscone can alleviate inflammatory pain induced by complete Freund's adjuvant (*Yu et al., 2020*). Previous studies have shown that proinflammatory cytokines, including IL-6, TNF-α, and IL-1β, play pivotal roles in inducing pain associated with the disease (*Kalpachidou et al., 2022*; *Zheng et al., 2024a*). Our observations indicated that muscone downregulated the expression levels of proinflammatory cytokines, including IL-1β, IL-6, TNF-α, IL-17, and

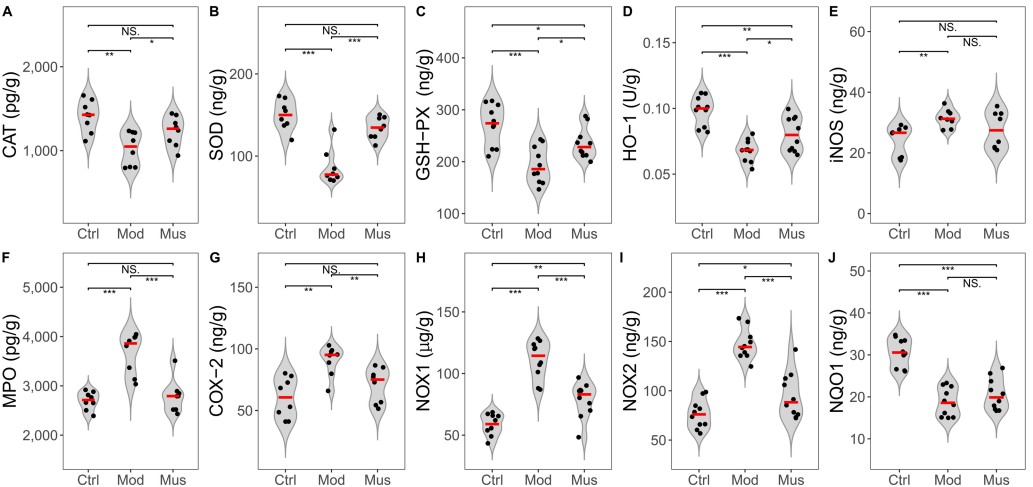

**Figure 8 Expression levels of ROS and antioxidases.** (A) CAT expression was significantly elevated in the muscone group compared with the model group. (B) SOD expression was significantly greater in the muscone group than in the model group. (C) GSH-PX expression was significantly greater in the muscone group compared with the model group. (D) HO-1 expression was significantly greater in the muscone group than in the model group. (E) iNOS expression in colon tissue was evaluated. (F) MPO expression was significantly lower in the muscone group than in the model group. (G) COX-2 expression was significantly lower in the muscone group than in the model group. (H) NOX1 expression was significantly lower in the muscone group than in the model group. (I) NOX2 expression was significantly lower in the muscone group than in the model group. (J) NQO1 expression in colon tissue was assessed. Ctrl, control group; Mod, model group; Mus, muscone group; NS, non-significant. Fresh colon tissue was harvested after the completion of the muscone treatment. Proteins associated with oxidative stress were quantified using ELISA after the dissociation of colon tissue. One-way ANOVA was used to compare the expression of ROS and antioxidases among the control group ($n = 8$ or 10), model group ($n = 8$ or 10), and muscone group ($n = 8$ or 10): * $p < 0.05$; ** $p < 0.01$, *** $p < 0.001$.

IL-33, in DSS-induced colitis (Fig. 7). Muscone may alleviate chronic pain associated with intestinal inflammation and improve the locomotor ability of mice with DSS-induced colitis.

Studies have demonstrated that a compound, Shexiang Huangqi Dropping Pills, a traditional Chinese medicine formulation mainly composed of musk, increases the concentrations of claudin, occludin, and ZO-1 in an *in vitro* blood–brain barrier model (*Li, Li & Zhang, 2022*). Musk maintains the stability of the prostate epithelial structure by regulating the expression levels of claudin, occludin, and ZO-1 (*Lin et al., 2015*). Claudin, occludin, and ZO-1 are crucial components of tight junctions, which maintain epithelial barrier function stability (*Schreiber et al., 2024*). We observed a significant increase in the levels of claudin, occludin, and ZO-1 in the mice treated with muscone (Fig. 6). Our results suggest that muscone treatment improves colon epithelial barrier function in DSS-induced colitis.

We observed a significant reduction in the expression levels of the proinflammatory cytokines IL-1β, IL-6, TNF-α, IL-17, and IL-33 in the mice treated with muscone compared with those in the model group (Fig. 7). Compared with those in untreated mice, the levels

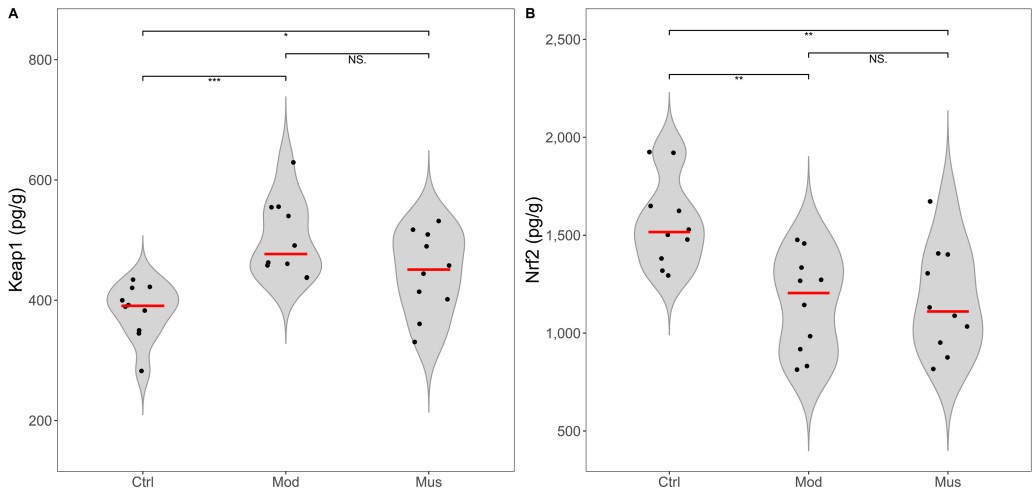

**Figure 9** **Expression levels of Keap 1 and Nrf2.** (A) Keap 1 expression in colon tissue. (B) Nrf2 expression in colon tissue. Ctrl, control group; Mod, model group; Mus, muscone group; NS, non-significant. Fresh colon tissue was harvested after the completion of the muscone treatment. Keap1 and Nrf2 proteins were quantified using ELISA after the dissociation of colon tissue. One-way ANOVA was used to compare the expression levels of key proteins among the control group ($n = 10$), model group ($n = 8$), and muscone group ($n = 10$): * $p < 0.05$; ** $p < 0.01$, *** $p < 0.001$.

of IL-4 and IL-10 were significantly elevated in mice treated with muscone. IL-4 and IL-10 are two important cytokines that promote anti-inflammatory processes (*Gul et al., 2023*). Many studies have demonstrated that muscone ameliorates diseases through its strong anti-inflammatory effects (*Wang et al., 2020*; *Liu et al., 2023*; *Yang et al., 2023*; *Li, Zhuang & Jiang, 2024c*). Muscone administration significantly reduces the inflammatory response following myocardial infarction and improves survival rates in mice (*Yu et al., 2020*). Collectively, our findings suggest that muscone exhibits potent anti-inflammatory activity in DSS-induced colitis.

Several studies have demonstrated that oxidative stress significantly contributes to tissue damage in IBD (*Bourgonje et al., 2020*; *Abd Elmaksoud et al., 2021*; *Lin et al., 2023*). DSS-induced colitis increases the expression level of MPO while decreasing the concentration of antioxidative enzymes (*Abd Elmaksoud et al., 2021*; *Muro et al., 2024*). Our observations revealed that the mice treated with muscone exhibited a significant upregulation of the expression of antioxidative enzymes, such as CAT, SOD, GSH-PX, and HO-1. Conversely, there was a significant downregulation of enzymes implicated in ROS production, including MPO, COX-2, NOX1, and NOX2 (Fig. 8). Previous research has demonstrated that muscone mitigates oxidative damage by inhibiting the generation of ROS to reduce oxidative stress (*Yu et al., 2014*; *Liu et al., 2022*). Furthermore, studies have shown that muscone significantly elevates the levels of SOD, HO-1, and GSH-PX, effectively counteracting oxidative stress and suppressing inflammatory responses (*Ma et al., 2013*; *Phung et al., 2020*; *Liu et al., 2022*; *Huang et al., 2025*). Our results confirmed that muscone has potent antioxidative properties. These findings indicate that muscone

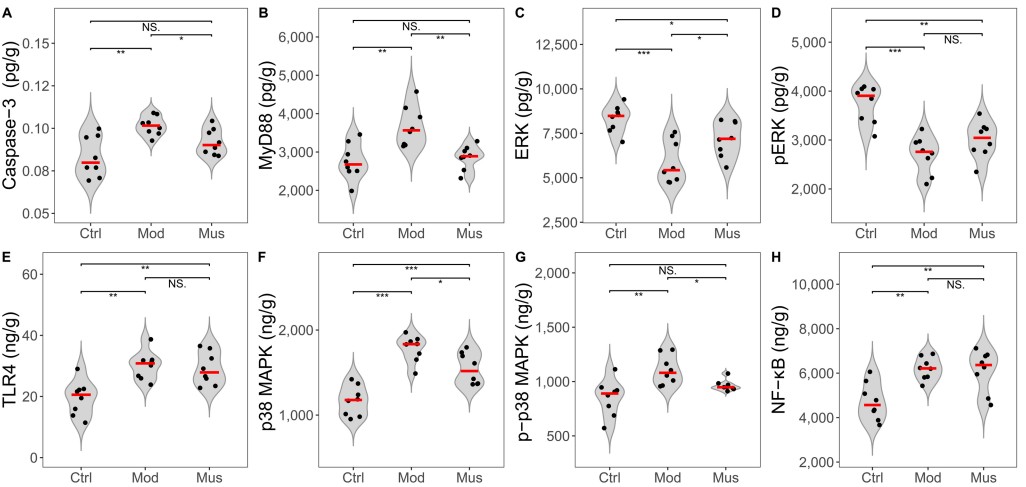

**Figure 10 Expression levels of key proteins in potential signalling pathways.** (A) Caspase-3 expression was significantly lower in the muscone group than in the model group. (B) MyD88 expression was significantly lower in the muscone group than in the model group. (C) ERK expression was significantly greater in the muscone group compared with the model group. (D) pERK expression in colon tissue was evaluated. (E) TLR4 expression in colon tissue was assessed. (F) p38 MAPK expression was significantly lower in the muscone group than in the model group. (G) p-p38 MAPK expression was significantly lower in the muscone group than in the model group. (H) NF-κB expression in colon tissue was measured. Ctrl, control group; Mod, model group; Mus, muscone group; NS, non-significant. Fresh colon tissue was harvested after the completion of the muscone treatment. Proteins associated with potential signalling pathways were quantified using ELISA after the dissociation of colon tissue. One-way ANOVA was used to compare the expression levels of key proteins among the control group ($n = 8$), model group ($n = 8$), and muscone group ($n = 8$), with significance levels indicated as follows: * $p < 0.05$; ** $p < 0.01$, *** $p < 0.001$.

may ameliorate IBD symptoms through scavenging reactive oxygen species, thereby demonstrating therapeutic potential.

Many studies have reported the role of the ERK signalling pathway in mediating anti-inflammatory processes that ameliorate DSS-induced colitis (*Qu et al., 2024*; *Su et al., 2024*). Our findings revealed a significant decrease in the ERK concentration and phosphorylation level in DSS-treated mice, which is consistent with the findings of previous studies (*Qu et al., 2024*). Research has shown that acetylcholine treatment activates the ERK signalling pathway *via* ERK phosphorylation to alleviate DSS-induced colitis (*Zheng et al., 2021*). However, we did not observe the anticipated increase in ERK phosphorylation reported in previous studies (Fig. 10) (*Zheng et al., 2021*). Our results indicate that the ERK signalling pathway may not be the primary mechanism by which muscone alleviates DSS-induced colitis.

MyD88 is a central node in anti-inflammatory, antioxidative, antiapoptotic, and tight junction regulation (*Baek et al., 2023*; *Miao et al., 2024*; *Zheng et al., 2024b*). The inhibition of MyD88 expression results in the downstream downregulation of MAPK, ROS, and apoptosis-related proteins (*Liu et al., 2018*; *Abdelzaher et al., 2023*; *Miao et al., 2024*; *Wei et al., 2024*; *Zheng et al., 2024b*). Studies have shown that MyD88 inhibition reduces damage and neuronal apoptosis in epilepsy (*Liu et al., 2018*). Caspase-3 is a proapoptotic protein

involved in activating the downstream apoptotic cascade (*Abdelzaher et al., 2023*; *Wei et al., 2024*). Studies have shown that trinitrobenzene sulfonic acid (TNBS)-induced colitis is ameliorated by inhibiting MyD88 expression, which subsequently downregulates downstream caspase-3 expression (*Miao et al., 2024*). Research has indicated that muscone enhances cell viability and inhibits apoptosis through the caspase-3 pathway (*Zhang et al., 2023*; *Sun et al., 2024*). Our findings revealed a significant increase in MyD88 and caspase-3 concentrations in DSS-induced colitis, which subsequently decreased after muscone treatment (Fig. 10). These findings align with previous research demonstrating that colitis can be ameliorated through the suppression of apoptosis *via* the MyD88 pathway modulation (*Miao et al., 2024*). ROS are considered primary factors in tissue destruction. The concentration of ROS significantly increases in DSS-induced colitis, whereas colitis is alleviated as ROS levels decrease (*Lin et al., 2023*). A significant decrease in ROS levels was observed following muscone treatment (Fig. 8), which is consistent with previous studies indicating that DSS-induced colitis can be mitigated through an ROS-scavenging strategy (*Lin et al., 2023*). The overproduction of ROS resulting from apoptosis is considered a primary trigger of programmed cell death (*Abdelzaher et al., 2023*). Apoptosis and increased ROS activate the MAPK signalling pathway, leading to an increase in inflammatory cytokines that exacerbate colitis (*Yue & López, 2020*; *Zheng et al., 2024b*). Our observations indicate that muscone treatment significantly reduces the levels of p38 MAPK and p-p38 MAPK, which are elevated in DSS-induced colitis (Fig. 10). The observed reductions in MAPK levels are consistent with previous studies demonstrating alleviation of colitis (*Yue & López, 2020*; *Zheng et al., 2024b*). Collectively, our findings suggest that muscone may alleviate DSS-induced colitis through the MyD88/p38 MAPK signalling pathway.

Our results have demonstrated the efficacy of muscone in treating IBD, but further studies are needed to confirm these findings. Our study only grouped subjects on the basis of a single dosage of muscone, which cannot comprehensively elucidate whether muscone treatment is dose dependent. Mucin-secreting goblet cells should be subjected to alcian blue staining to evaluate their distribution *via* histological studies. Detection of inflammatory cytokines and proteins related to oxidative stress requires additional protein expression analyses, such as western blot, immunohistochemistry, or immunofluorescence, to ensure reproducibility of the results. Further investigations into cytokine-producing cells, particularly regulatory T cells (Tregs), are essential to validate these results. Moreover, additional studies on key signalling pathway proteins, such as MyD88 and MAPK, influenced by muscone, are necessary to determine the reliability of these findings.

## CONCLUSIONS

Our study reports the therapeutic effects of muscone on DSS-induced colitis. Muscone decreased the histopathological damage score, increased the ratio of colon length to body weight, and improved the survival probability. Treatment with muscone significantly improved the gait characteristics of mice with DSS-induced colitis. Muscone treatment downregulated the expression of the proinflammatory cytokines IL-1β, IL-6, IL-17,

IL-33, and TNF-α while increasing IL-4 and IL-10 expression. Muscone has significant antioxidative properties that can reduce various types of ROS. Collectively, muscone has been shown to alleviate symptoms associated with DSS-induced colitis, potentially through the inhibition of the MyD88/p38 MAPK signalling pathway. Our findings suggest that muscone could be a promising candidate for treating IBD in clinical studies. However, further validation of our findings is necessary, and the underlying mechanism of action of mucone in DSS-induced colitis requires elucidation through comprehensive future studies.

## ACKNOWLEDGEMENTS

The authors express their gratitude to Zhidong Shen, Wen Zhang, Can Zhang, and all other contributors to the study.

### Funding

This work was supported by the Guizhou Provincial Basic Research Program (Natural Science) (No. Qian ke he ji chu [2020]1Y386), the Academic Project for New Researchers of Guizhou University of Traditional Chinese Medicine (No. Gui ke he xue shu xin miao [2023]-55), and the Research initiation fund of Guizhou University of Traditional Chinese Medicine. The funders had no role in study design, data collection and analysis, decision to publish, or preparation of the manuscript.

### Grant Disclosures

The following grant information was disclosed by the authors:
The Guizhou Provincial Basic Research Program (Natural Science): No. Qian ke he ji chu [2020]1Y386.
The Academic Project for New Researchers of Guizhou University of Traditional Chinese Medicine: No. Gui ke he xue shu xin miao [2023]-55.
The Research initiation fund of Guizhou University of Traditional Chinese Medicine.

### Competing Interests

The authors declare there are no competing interests.

### Author Contributions

- Gang Yao conceived and designed the experiments, performed the experiments, analyzed the data, prepared figures and/or tables, authored or reviewed drafts of the article, and approved the final draft.
- Jian Zhang performed the experiments, analyzed the data, prepared figures and/or tables, authored or reviewed drafts of the article, and approved the final draft.
- Lingyan Zhang analyzed the data, prepared figures and/or tables, authored or reviewed drafts of the article, and approved the final draft.
- Hai Zhao performed the experiments, analyzed the data, prepared figures and/or tables, authored or reviewed drafts of the article, and approved the final draft.

- Shuguang Wu analyzed the data, authored or reviewed drafts of the article, and approved the final draft.
- Hongmei Yang analyzed the data, authored or reviewed drafts of the article, and approved the final draft.
- Jiangwei Yu analyzed the data, authored or reviewed drafts of the article, and approved the final draft.

## Animal Ethics

The following information was supplied relating to ethical approvals (*i.e.*, approving body and any reference numbers):

The Animal Experimental Ethical Inspection of Guizhou University of Traditional Chinese Medicine provided full approval for this research (No. 20230062).

## Data Availability

The raw measurements are available in the Supplementary Files.

## Supplemental Information

Supplemental information for this article can be found online at http://dx.doi.org/10.7717/peerj.19397#supplemental-information.

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
