# Peer review of "Alleviation of dextran sulfate sodium (DSS)-induced colitis in mice through the antioxidative effects of muscone via the MyD88/p38 MAPK signalling pathway"

_PeerJ, doi:10.7717/peerj.19397_

## Round 0.1 · original submission · Major Revisions

The reviewers have highlighted numerous aspects of the experimental design and reporting, warranting a major revision of your manuscript. Please ensure you provide a comprehensive reply and correction to all requests.

·

Basic reporting

Manuscript ID 102624v1
This paper is related to reviewing the manuscript titled " The alleviation of muscone on DSS-induced ulcerative colitis in mice through its antioxidative effects via the MyD88/ERK signaling pathway"
This work employed muscone to treat DSS-induced colitis in C57BL/6 mice. Muscone therapy significantly decreased gross bleeding and histological damage. Gait study showed improvements in swing time, brake time, propel time, and posture.

Experimental design

Firstly, Although the proposed study is successful in terms of organization, presentation, content and results, major revision given in the following items need to be performed.
1) Improve the abstract and conclusion section, enhance the manuscript to convey the purpose, objectives, method and major findings, especially results.
2) Use abbreviations after the first use in the text, in the abstract and throughout the paper.
3) The literature review is quite insufficient in the introduction section. Complete the introduction and literature sections of the manuscript by providing similar studies from the years 2023-2024 and/or new and current studies that will attract the attention of the readers.
4) Are there any other statistical and/or learning-based methods used by the authors other than the ANOVA test using R programming? Why are their methods weak?
5) More analysis results should be included in the results and findings section.
6) A comprehensive table is needed to present the results.
7) The conclusion section really needs to be improved
8) Clean the paper of English spelling and punctuation errors

Validity of the findings

As above

Reviewer 2 ·

Basic reporting

Some comments to the figures and figure legends.

1. The consistency of showing the groups in the figures, especially the order of the groups. In Figure 1 (a) (b) (c) (f), the authors showed Control - Model - Muscone, which is the best order, but in Figure 2-6 are showing Model - Control - Muscone. I'd suggest to keep the order showing in Figure 1. Inconsistency in Figure 1 (d) (e), please revise properly.

2. For the Figure Legends, please have a title of the figure describing the finding of the figures. After the title, please summarize how the experiments were processed, i.e. the source of the samples, the time point collecting the samples, the methods for bioanalysis of the cytokines/chemokines/proteins, etc. In the end, please show the statistical analysis, mean+/-SD or SEM, the number of animal or independent experiments in each group, and the p-value for defining the statistical significance.

3. Some comments on Figure Legends:
(1) Figure 1, only (a) - (g) were mentioned, but not (h) and (i)
(2) Figure 1, please clearly label the groups for (g) (h) and (i).
(3) Figure 1 (a) (b) (c) (d) (e) (f), please show the significant difference between groups.
(4) Figure 2-6, please mention which statistical analysis were used in the figure legends.

Experimental design

Several comments on the experimental design:

1. For the survival probability, only 16 days were monitored, was there any reason to stop the study or the humane concern after 16 days? If it's the time point to euthanize the mice on day 16 for tissue collections, please explain the scientific approach.

2. Please clearly describe the timepoint for collecting the tissues, i.e. how many days after treatment, how to process the tissues, and the timepoints to collect the samples for ELISA and gene expression.

Validity of the findings

Several comments to the study design and the results.

1. In Figure 1 (e), muscone did not significantly increase the colon length, which is the most important clinical observation in the DSS-induced colitis model. Please explain the possible reason that did not correlate to the significant difference observed in histopathological damages/scores in Figure 1 (d).

2. In Figure 2, there are some observations that in (d) Time Stance, (e) Time of Stride, and (g) Stride Frequency, LF and RF showed different results. Is this observation commonly happen in behavior evaluation? Or any possibility cause this difference?

3. For the expressions of epithelial barrier proteins, only increased occludin was observed, but not in claudin1, ZO-1, and MUC-2, what's the possible reason only occludin was affected and improved by muscone but not other epithelial barrier proteins? Did the increased occludin relief the severity of DSS-induced colitis?

4. For the inflammatory cytokines shown in Figure 4, what did the samples come from? Were they from the supernatants of primary tissue culture or directly from the tissue dissociation? If from the tissue dissociation, how did the author normalize the cytokine levels?

5. For the inflammatory cytokines, in Figure 4 (a) IL-1β, Model and Muscone groups showed significant difference with small change but no significant difference in (b) IL-6, please confirm the statistical analysis.

6. Did the author analyze the immuno-phenotyping of T cell populations in intestine and colon, such as Treg, Th1, Th2, and Th17 cells? Treg is the subtype of CD4+ T cells in DSS-induced colitis model, which the cells contributing on IL-10 productions in intestine and colon.

7. In Figure 6, muscone decreased MyD88 expression but increased ERK expression and no effects on NF-κB expression. What's the possible regulation causing different trends on MyD88, ERK and pERK, and NF-κB expression, whereas ERK and NF-κB are the downstream signaling of MyD88. Please add into the discussion section.

---

## Round 0.2 · Minor Revisions

Please address all comments provided by the reviewer and submit point-wise responses along with a revised manuscript.

**Language Note:** The review process has identified that the English language must be improved. PeerJ can provide language editing services - please contact us at [email protected] for pricing (be sure to provide your manuscript number and title). Alternatively, you should make your own arrangements to improve the language quality and provide details in your response letter. – PeerJ Staff

Reviewer 2 ·

Basic reporting

Comment in the section "Additional comments".

Experimental design

Comment in the section "Additional comments".

Validity of the findings

Comment in the section "Additional comments".

Additional comments

1. The changes, especially the line numbers, mentioned in the rebuttals and revised manuscript don't match. It's really difficult to read.
2. The revised discussion only mentioned the results of the previous studies but not being discussed and compared to the results from the present study.
3. The manuscript needs a major revision on scientific writing.

---

## Round 0.3 · accepted · Accept

Authors have addressed all of the reviewers' comments. I have assessed the revision myself, and I am happy with the current version. Manuscript is ready for the publication.